# A systematic review to identify research gaps in studies modeling MenB vaccinations against *Neisseria* infections

**Soeren Metelmann**[1]*, **Alexander Thompson**[2], **Anna Donten**[2], **Segun Oke**[3], **Suzy Sun**[4], **Ray Borrow**[5], **Feng Xu**[3], **Roberto Vivancos**[1], **Valerie Decraene**[1], **Lorenzo Pellis**[3], **Ian Hall**[3]

**1** Field Service, UK Health Security Agency, Liverpool, United Kingdom, **2** School of Health Sciences, Manchester University, Manchester, United Kingdom, **3** School of Mathematics, Manchester University, Manchester, United Kingdom, **4** Blood Safety, Hepatitis, Sexually Transmitted Infections and HIV Division, UKHSA, London, United Kingdom, **5** Meningococcal Reference Unit, UK Health Security Agency, Manchester Royal Infirmary, Manchester, United Kingdom

* soeren.metelmann@ukhsa.gov.uk

## Abstract

The genus *Neisseria* includes two major human pathogens: *N. meningitidis* causing bacterial meningitis/septicemia and *N. gonorrhoeae* causing gonorrhoea. Mathematical models have been used to simulate their transmission and control strategies, and the recent observation of a meningococcal B (MenB) vaccine being partially effective against gonorrhoea has led to an increased modeling interest. Here we conducted a systematic review of the literature, focusing on studies that model vaccination strategies with MenB vaccines against *Neisseria* incidence and antimicrobial resistance. Using journal, preprint, and grey literature repositories, we identified 52 studies that we reviewed for validity, model approaches and assumptions. Most studies showed a good quality of evidence, and the variety of approaches along with their different modeling angles, was assuring especially for gonorrhoea studies. We identified options for future research, including the combination of both meningococcal and gonococcal infections in studies to have better estimates for vaccine benefits, and the spill over of gonorrhoea infections from the heterosexual to the MSM community and vice versa. Cost-effectiveness studies looking at at-risk and the wider populations can then be used to inform vaccine policies on gonorrhoea, as they have for meningococcal disease.

## 1. Introduction

*Neisseria gonorrhoeae* and *Neisseria meningitidis* are closely related bacteria that cause a significant global burden of disease. While vaccines are licensed and routinely used for *N. meningitidis*, no vaccine is licensed for *N. gonorrhoeae*. In addition, control of gonorrhoea is becoming increasingly difficult due to widespread antimicrobial resistance (AMR).

But there is hope: meningococcal vaccines potentially offer some cross protection against gonorrhoea [1, 2]. Recent observations and retrospective studies from Cuba [3], New Zealand

**Data Availability Statement:** All relevant data are within the manuscript and its Supporting Information files.

**Funding:** This work was supported by the Wellcome Trust under grant "Impact of vaccines on antimicrobial resistance" [219792/Z/19/Z]. The funders had no role in study design, data collection and analysis, decision to publish, or preparation of the manuscript.

**Competing interests:** RB performs contract research on behalf of UKHSA for GSK, Pfizer, and Sanofi. RV has received research funding on behalf of Public Health England from GSK and Gilead Sciences in the past. This does not alter our adherence to PLOS ONE policies on sharing data and materials. All other authors have declared that no competing interests exist.

[4], Canada [5], USA [6–8], and Australia [9] reported between 31% and 59% reduction in incidence rates of gonorrhoea in those vaccinated with meningococcal B (MenB) outer membrane vesicle (OMV) containing vaccines. This is because minor antigens in the OMV and a *Neisseria* heparin binding protein in other MenB vaccines are also surface exposed in *N. gonorrhoeae* [10].

The UK introduced a MenB vaccine (Bexsero, GSK) into the national infant immunization schedule from 2015 [11]. Cost-effectiveness of the MenB vaccine against meningococcal disease in adolescents in the UK is borderline given the low impact against carriage acquisition and thus no indirect protection [12], the relatively low incidence of *N. meningitidis* group B infections and the cost of the vaccine [13]; hence immunization in the UK has been targeted to infants and direct protection. For this to have any effect on the incidence of gonococcal infections it will take another 10 to 15 years if the effect, diluted by waning immunity, is noticeable at all.

Given the uncertainty around the effectiveness and duration of this potential vaccine, and around the best vaccination age and population, mathematical models can be a useful tool to simulate different scenarios and strategies. They can explore the impact of vaccines with different characteristics on the long-term gonorrhoea incidence and the level of AMR. If linked with health economics, these models can also advise on the cost-effectiveness of different vaccination strategies.

And while there has been a growing number of modeling studies on the use of MenB vaccines against gonorrhoea, especially since the NZ study in 2017, there has not been a systematic review of the modeling literature so far. Here we searched a range of scientific and grey literature databases and summarized results to give an overview of the different techniques that have been used to model MenB vaccination scenarios for *Neisseria* infections, both gonococcal and meningococcal. A secondary aim was to summaries how the spread of AMR in *Neisseria sp.* was modeled and what impact vaccination campaigns could have on the spread of AMR. This review seeks to identify existing research gaps in this field.

## 2. Methods

This systematic review was conducted and written up following the Preferred Reporting Items for Systematic reviews and Meta-Analyses (PRISMA) 2020 guidelines [14], and a PRISMA checklist is available in S4 File. The review protocol was not registered prospectively though, it is available in S5 File.

### 2.1 Search strategy

The inclusion and exclusion criteria were developed following the Population, Intervention, Comparison, Outcomes, and Study (PICOS) framework [15], see Table 1.

We applied the following inclusion criteria:

1. Title or abstract had to mention a mathematical model with bacterial transmission mechanisms

2. Title or abstract had to mention either or both of the following infections:

    a. Gonococcal infection (gonorrhoea)

    b. Serogroup B meningococcal infection (bacterial meningitis/septicemia)

3. Title or abstract had to mention one or both of the following subjects:

    a. Antimicrobial resistance

    b. Vaccination (with Trumenba, Bexsero, or other MenB vaccines)

**Table 1. Criteria for study inclusion following the PICOS framework.**

| | |
|---|---|
| Participants or Population | This review will consider all studies that involve persons eligible for MenB vaccination |
| Interventions | Interventions of interest included those related to the following: |
| | Effectiveness and/or efficacy of MenB Vaccine |
| | Continuation of existing vaccination programmes |
| | Screening systems |
| | Assessment strategies of medication |
| | Intervention programmes |
| | Specific clinical interventions |
| Comparisons | Infection and AMR levels in targeted groups at greater risk of gonococcal infections with and without vaccination |
| Outcome of Interest | A transmission model at population scale of GC and MenB infection for the UK |
| | Cost-effective vaccination strategies to reduce MenB and GC infection incidence and AMR. |
| | Simulated planned activities for vaccine strategies using the transmission dynamic model of GC and MenB infection and vaccination |
| Study designs | Modeling study using direct or indirect measurement methods to evaluate the effectiveness or efficacy of interventions/strategies relating to gonococcal or meningococcal infections, and the impact on AMR in this infection. |

Abbreviations: MenB–serogroup B meningococcal, AMR–antimicrobial resistance, GC–gonococcal

We applied the following exclusion criteria:

1. All non-primary studies such as reviews or meta-analyses unless they used or published new data

2. Studies that were irretrievable or conference abstracts for oral talks

3. Studies not available in English

4. Studies that featured the search terms but only for definitions, descriptions, or referred to for comparison, and that do not answer the research question of this study.

5. Studies reporting genomic sequencing models, within-host models, conventional statistical modeling or analysis, and models without between-host transmission mechanisms

6. Studies on meningococcal conjugate vaccine that are not serogroup B (such as modeling studies for the African meningitidis belt).

## 2.2 Data sources

We searched the journal databases MEDLINE and EMBASE via OVID, PubMed, and Scopus. We searched for preprint articles on OSF Preprints (incl. aRxiv and bioRxiv), and on medRxiv via a google scholar search. Finally, we also searched for grey literature including conference publications, technical reports, dissertations etc. on the repositories base-search.net, British Library, and OpenGrey. The search was conducted on 30[th] June 2023 for all databases, and the search strings used for databases and repositories are available in the protocol in S5 File. A list of all identified studies is found in S7 File.

## 2.3 Screening process

All search results were screened to eliminate duplicate entries, including preprints that were later published as journal articles. After deduplication, titles and abstracts were screened for

our inclusion and exclusion criteria as defined above. If the screening of the title and abstract was inconclusive, the whole paper was screened to make sure the selection criteria could be applied correctly. This screening process was done by two authors independently and the results were compared. If the two authors came to a different conclusion for a particular study, the study's abstract and full text was discussed in detail until an agreement was reached. If there was no agreement, a third author acted as reviewer to arbitrate a final decision.

## 2.4 Data synthesis

A qualitative synthesis of the included studies was used to organize the modeling studies. An extraction form was developed based on the following categories: study title, infectious disease system, model type, model formulation/class, transmission route, methodology, validation technique, intervention target, type of data used, and health economic analysis. Data was extracted and organized by three different authors, depending on their expertise. The data extraction template is available in S2 File.

A descriptive analysis of the data generated from the systematic search, in line with the study protocol, is reported using flow charts to illustrate included and excluded publications and their sources and tables (to present studies, models, and setting characteristics). The main model assumptions, including model structure, setting, vaccines, AMR, and health economics, are summarized for meningococcal and gonococcal studies separately.

## 2.5 Quality assessment

We use the standardized survey from Lo et al. [16] to assess the quality of evidence and the studies' usefulness for decision making. The modeling studies were assessed by checking

1. Model structure and assumptions

2. Model calibration

3. Influential model inputs

4. Robustness of sensitivity analysis

5. Robustness of uncertainty analysis

6. Face validity

7. External or internal validation

8. Generalizability

9. Funder conflict of interest

We have assigned a "+" for each category if the study ticked all or most of the category's checks. A list of the checks can be found in S3 File. All studies were included in the review regardless of their validity rating.

## 3. Results

### 3.1 Selection process

We found a total of 479 documents with online search engines and an additional 2 documents were identified through further reading. Of the 479 documents, 306 were identified as duplicates, either having been found by multiple search engines or being preprints or thesis chapters that were later published as journal articles. A further 4 documents were not retrievable in English and thus excluded. The remaining 169 documents were then checked for inclusion

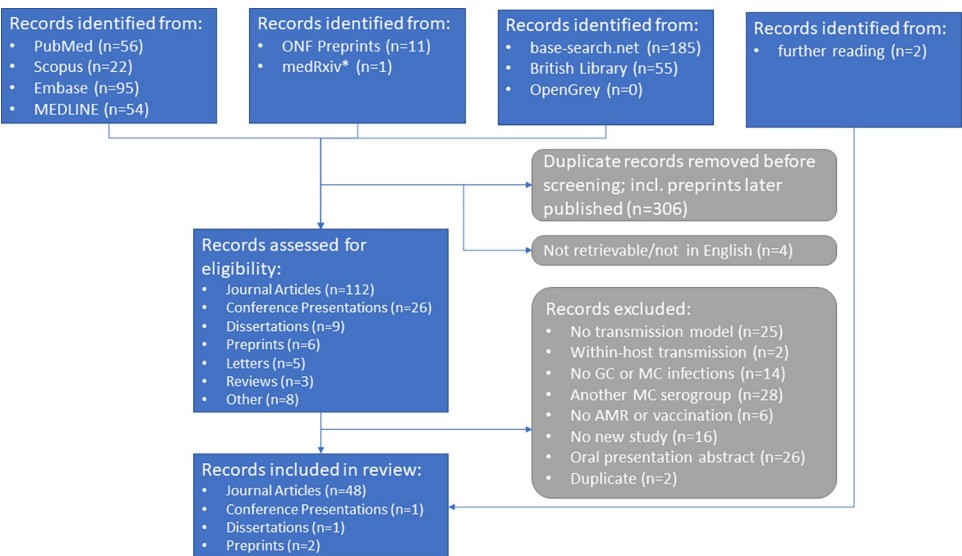

**Fig 1. PRISMA flowchart.** Literature search, screening, and inclusion process. *medRxiv searched via google scholar using "source:medRxiv".

and exclusion criteria and a total of 52 documents were eligible for full text review. See Fig 1 for the process and reasons for exclusion. The 52 included documents comprise 48 journal articles, 1 dissertation, 2 preprints, and 1 conference article.

## 3.2 Study characteristics

After some initial modeling of AMR in GC [17, 18], the importance of the rise in AMR and vaccination strategies as a possible solution have only been analyzed from 2012 onwards, see Fig 2. In the 10-year period from 2013 to 2022, an average of 4 articles have been published per year. In addition, three topical dissertations and 25 conference abstracts have been found for this period (bearing in mind that abstracts have probably been sparsely archived before 2010). Study characteristics are presented in Table 2. The 52 included modeling studies were describing either gonococcal infection (32) or meningococcal infections (20) but not both, see S1 Fig in S1 File.

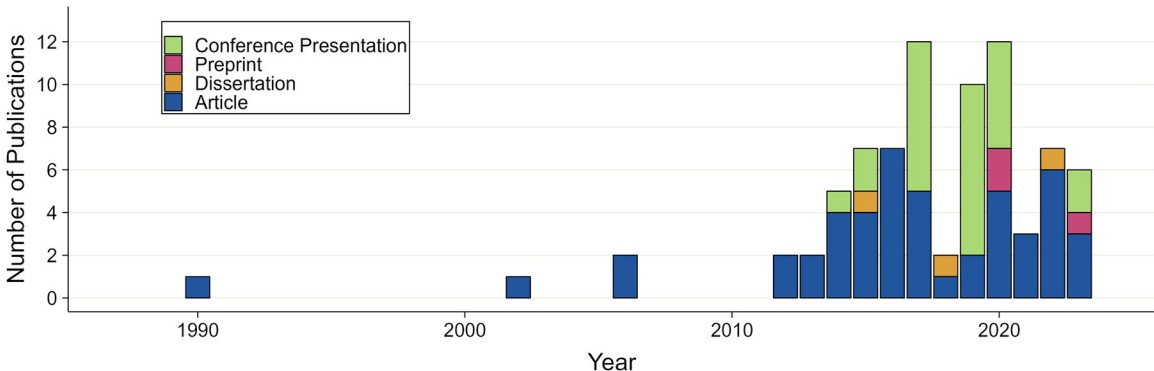

**Fig 2. Publication history.** Timeline of publications matching our inclusion criteria, underlying data available in S3 Table (S1 File).

**Table 2. Characteristics of included studies.**

| | Study | Research Question | Inf | Population | Country | Model | AMR | Vaccine | Assessment |
|---|---|---|---|---|---|---|---|---|---|
| 1 | Argante et al., 2016 [28] | How can the effectiveness of meningitis immunization campaigns be quantified in real time? | MC | General | England | Markov | - | Bexsero | +++++++ |
| 2 | Beck et al., 2020 [26] | How can MenB and MenACWY vaccines strategies decrease IMD? | MC | General | England | ODE | - | Bexsero | +++++++ |
| 3 | Bos et al., 2006 [38] | Is a combined MenB x pneumococcal vaccine cost effective? | MC | General | Netherlands | Markov | - | hypothetical | +++++++ |
| 4 | Breton et al., 2020 [36] | Is a MenB-FHbp vaccine cost effective? | MC | General | Canada | Markov | - | Trumenba | +++++ |
| 5 | Buyze et al., 2018 [43] | Do regular screenings of the population lower GC prevalence or just contribute to the rise in AMR? | GC | MSM | Belgium | Markov Network | Hypothetical AB | - | +++ |
| 6 | Carey et al., 2022 [69] | What impact would a MC vaccine have on GC prevalence? | GC | Hetero-HL | USA | ODE | - | Hypothetical | +++++++ |
| 7 | Chan et al., 2012 [64] | How much does AMR contribute to the current rise in GC cases and what are best treatment strategies? | GC | General-HL | Canada, USA | ODE | 2 hypothetical Abs | - | +++++ |
| 8 | Christensen et al., 2013 [13] | Is it cost effective to have universal MenB vaccination? | MC | General | England | Markov, ODE | - | hypothetical | ++++++++ |
| 9 | Christensen et al., 2014 [20] | Is it cost effective to have universal MenB vaccination? | MC | General | England | ODE | - | Bexsero | ++++ |
| 10 | Christensen et al., 2016 [31] | Is it cost effective to have universal MenB vaccination? | MC | General | Germany | Markov, ODE | - | Bexsero | +++++++ |
| 11 | Christensen & Trotter, 2017 [37] | Is it cost effective to have catch-up MenB vaccinations? | MC | General | England | ODE | - | Bexsero | +++ |
| 12 | Chung et al., 2020 [23] | Is it cost effective to have MenB vaccinations for college students? | MC | College students | USA | Markov | - | Bexsero, Trumenba | ++++++++ |
| 13 | Craig et al., 2015 [61] | What impact would a hypothetical vaccine have on GC prevalence? | GC | Hetero-HL | Australia | Agent-based Markov | - | hypothetical | ++++++ |
| 14 | Duan et al., 2021 [42] | What test and treat strategies could eliminate imported GC strains? | GC | MSM, 16-65yo | Australia | Markov Network | Ceftriaxone | - | +++++++ |
| 15 | Fingerhuth et al., 2016 [47] | Does more treatment lead to more AMR? | GC | Hetero-HL, MSM-HL | hypothetical (data from UK, USA) | ODE | Ciprofloxacin, Cefixime | - | ++++++++ |
| 16 | Fingerhuth et al., 2017 [48] | How can point-of-care testing tackle AMR? | GC | Hetero-HL, MSM-HL | hypothetical | ODE | Hypothetical AB | - | +++++++ |
| 17 | Gasparini et al., 2016 [35] | Is it cost effective to have universal MenB vaccination? | MC | General | Italy | Markov | - | Bexsero | +++++++ |
| 18 | Graňa et al., 2021 [29] | How can MenB and MenACWY vaccines strategies decrease IMD? | MC | General | Chile | ODE | - | Bexsero | +++ |
| 19 | Handel et al., 2006 [51] | How can mutations compensate for fitness loss that comes with AMR? | GC | General high risk, 15-39yo | hypothetical | ODE | Hypothetical AB | - | ++++++ |

*(Continued)*

**Table 2.** (Continued)

| | Study | Research Question | Inf | Population | Country | Model | AMR | Vaccine | Assessment |
|---|---|---|---|---|---|---|---|---|---|
| 20 | Heijne et al., 2020 [52] | What impact would a vaccine have on GC transmission and AMR? | GC | MSM-HL, 15-60yo | Netherlands | ODE | Ceftriaxone | MeNZB | ++++++ |
| 21 | Hogea et al., 2016 [27] | Would vaccination against MenB lead to serogroup replacement? | MC | General | UK, Czech Republic | PDE | - | Bexsero | ++++ |
| 22 | Huang et al., 2022 [24] | What impact would a MenABCWY pentavalent vaccine have on IMD? | MC | General | USA | Markov | - | hypothetical | +++ |
| 23 | Huels et al., 2014 [33] | What impact would a vaccine have on MC incidence? | MC | General | UK | ODE | - | Bexsero | ++++++ |
| 24 | Hui et al., 2015 [71] | How can point-of-care testing tackle AMR? | GC | Indigenous, 15-35yo | Australia | Agent-based Markov | Ciprofloxacin | - | +++ |
| 25 | Hui et al., 2017 [44] | What impact have imported cases on GC prevalence? | GC | MSM | Australia | Agent-based Markov | Ciprofloxacin | - | +++ |
| 26 | Hui et al., 2022 [45] | What impact would a hypothetical vaccine have on GC prevalence? | GC | MSM-HL, 16-80yo | Australia | Agent-based Markov | - | hypothetical | ++++ |
| 27 | Kreisel et al., 2021 [72] | What is true GC prevalence, incidence, and AMR proportion? | GC | General, 15-39yo | USA | ODE | Ceftriaxone, Cefixime, Azithromycin, Ciprofloxacin, Penicillin, Tetracycline | - | +++++++ |
| 28 | Landa et al., 2017 [73] | What impact do different modeling techniques have on vaccine cost effectiveness results? | MC | General | Italy | Markov | - | hypothetical | +++ |
| 29 | Lecocq et al., 2016 [22] | Is it cost effective to have universal MenB vaccination? | MC | General | France | Markov | - | Bexsero | +++++++ |
| 30 | Looker et al., 2023 [57] | What impact would adolescent GC vaccination have? | GC | Hetero-HL | England | ODE | - | Bexsero | +++++++ |
| 31 | Padeniya, 2022 [74] | What impact would a vaccine have on GC prevalence in FSW? | GC | Sex workers and clients | Australia | ODE | - | MeNZB | +++++++ |
| 32 | Pinsky & Shonkwiler 1990 [17] | What equilibria can a model with AMR and AMS strains have? | GC | General-HL | hypothetical | ODE | Penicillin | - | +++ |
| 33 | Pouwels et al., 2013 [34] | Is it cost effective to have universal MenB vaccination? | MC | General | Netherlands | Markov | - | Bexsero | +++++++ |
| 34 | Régnier & Huels, 2014 [68] | Could it be cost effective to use MenB vaccination against GC? | GC | General | USA | Markov | - | Bexsero | +++++++ |
| 35 | Reichert et al., 2023 [66] | What strategy should be used to introduce a novel antibiotic against GC? | GC | MSM-HL | US | ODE | Ceftriaxone, Hypothetical AB | - | +++++++ |
| 36 | Riou et al., 2023 [49] | How can the spread of antibiotic resistance in GC be modeled? | GC | Hetero, MSM | UK | ODE | Ciprofloxacin, Azithromycin, Cefixime, Ceftriaxone | - | ++++++++ |
| 37 | Scholz et al., 2022 [32] | Is it cost effective to have universal MenB vaccination? | MC | General | Germany | ODE | - | Bexsero | ++++++ |
| 38 | Simpson & Roberts, 2012 [25] | What impact did a vaccination campaign have on MC incidence? | MC | General | New Zealand | ODE | - | hypothetical | +++++ |

(*Continued*)

**Table 2.** (Continued)

| | Study | Research Question | Inf | Population | Country | Model | AMR | Vaccine | Assessment |
|---|---|---|---|---|---|---|---|---|---|
| 39 | Trecker et al., 2015 [67] | How do different model techniques affect results on AMR elimination? | GC | General-HL | hypothetical | ODE | Hypothetical AB | - | ++++ |
| 40 | Tsoumanis et al., 2023 [46] | How are screening frequency and development of AMR in GC linked? | GC | MSM-HL | Belgium | Markov network | Azithromycin, Ceftriaxone | - | +++++++ |
| 41 | Tu et al., 2014 [21] | Is it cost effective to have universal MenB vaccination? | MC | General | Canada | Markov | - | Bexsero | +++++++ |
| 42 | Tuite et al., 2017 [65] | How can point-of-care testing tackle AMR? | GC | MSM-HL | USA | ODE | Ciprofloxacin, Azithromycin, Ceftriaxone | - | +++++++ |
| 43 | Turner & Garnett, 2002 [18] | What impact does the timing of treatment have on competing strains in an outbreak? | GC | General-HL | hypothetical | ODE | Hypothetical AB | - | ++++ |
| 44 | Whittles et al., 2017 [58] | What are fitness costs associated with AMR? | GC | MSM | England | Markov | Cefixime | - | ++++++++ |
| 45 | Whittles et al., 2019 [60] | Can dynamic network models better reflect transmission? | GC | MSM-HL, 16-74yo | England | Markov network | - | hypothetical | +++++++ |
| 46 | Whittles et al., 2020 [59] | What impact would a vaccine have on GC transmission and AMR? | GC | MSM-HL | England | Markov | Hypothetical AB | MeNZB | +++++++ |
| 47 | Whittles et al., 2022 [70] | Could it be cost effective to have risk group specific vaccinations? | GC | MSM-HL | England | ODE | - | hypothetical | +++++++ |
| 48 | Xiridou et al., 2015 [53] | What treatment strategies can prevent an increase in AMR? | GC | MSM-HL | Netherlands | ODE | 3 hypothetical Abs | - | +++++++ |
| 49 | Xiridou et al., 2016 [54] | Is dual therapy more cost effective compared to monotherapy? | GC | MSM-HL | Netherlands | ODE | 2 hypothetical Abs | - | ++++++++ |
| 50 | Yaesoubi et al., 2020 [55] | How can different surveillance strategies prolong the use of antibiotics? | GC | MSM | USA | Markov | 3 hypothetical Abs | - | ++++++++ |
| 51 | Yaesoubi et al., 2022 [56] | Can local AMR-thresholds prolong the use of antibiotics? | GC | MSM | USA (16 cities) | Markov | 3 hypothetical Abs | - | ++++++++ + |
| 52 | Zienkiewicz et al., 2019 [63] | How can point-of-care testing tackle AMR? | GC | London MSM | England | Agent-based Markov | Ciprofloxacin, Ceftriaxone | - | +++++++ |

## 3.3 Meningococcal (MC) studies

**3.3.1 Model approaches.** The MC models are either deterministic differential equation models or stochastic Markov models. Ordinary differential equation (ODE) models are widely used for modelling dynamic transmission of diseases because they are simpler to implement and interpret with a single deterministic outcome. Markov models account for random variation in their inputs and yield outcome probabilities, thus illustrating the uncertainty in the process. The work on meningococcal serogroup C by Trotter et al. [19] has often been cited by studies of both model types, see Fig 3, and Trotter's group have used both differential equation and Markov models to analyze MC transmission later on [13, 20]. Nearly all models simulated dynamic disease transmission that can reflect non-linear effects, only some models assume a constant force of infection year on year and thus do not have dynamic transmission [21, 22].

**3.3.2 Model structure.** As all studies included some sort of vaccination, they mostly followed a SIRS (susceptible-infected-recovered/vaccinated-susceptible) structure, in which an

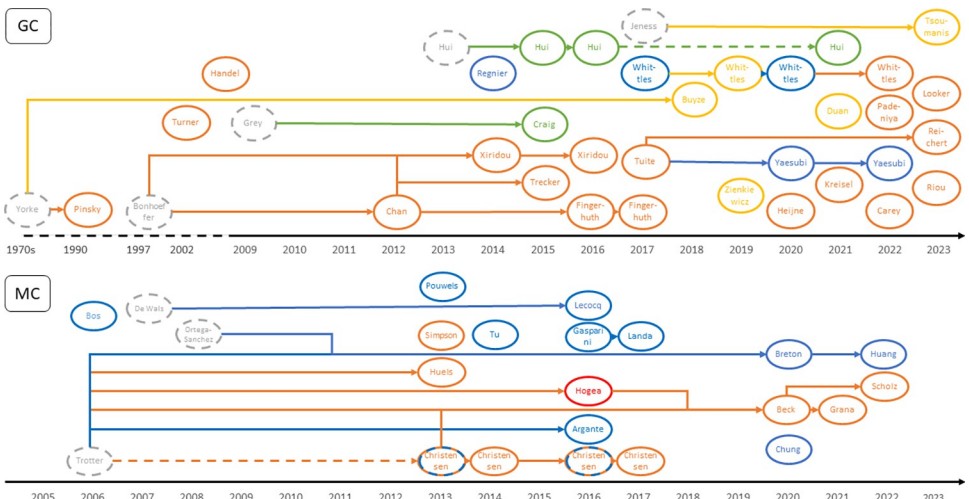

**Fig 3.** Phylogeny of *N. gonorrhoeae* (top) and *N. meningitidis* (bottom) models, indicated by first author. The colors denote the different model types: ODE (orange), PDE (red), Network (yellow), Population-based Markov (blue), Individual-based Markov (green). Studies in grey are not included in this review.

immune state was reached after infection or vaccination. With waning immunity over time, people return to the susceptible population. In some cases, an additional non-symptomatic but infectious exposed state was used [23–25], other models used additional infection classes for multiple meningococcal strains [26, 27].

All of the models either split the population into age classes and used age-dependent contact matrices for bacterial transmission (e.g. [13, 28]) to account for the age heterogeneity in meningococcal incidence, or they only looked at a single specific age group [23].

**3.3.3 Setting and population.** Almost all studies focus on developed countries only (Western Europe, USA and Canada, Australia and New Zealand), with the exception of one study set in Chile [29]. This is in part a result of our inclusion criteria, as many studies model transmission of meningococcal serogroups A, C, or W in the African meningitis belt. However, we wanted to exclude these studies because of their different setting and effective vaccination strategies that are already in place [30]. The MC studies looked at younger age groups like infants, adolescents, college students or analyzed how vaccination programmes in these younger age groups affected the general population.

The models were all parameterized with national population and infection data to different degrees. Most calibrated their model parameters to point estimate or time series data of national incidence or prevalence [13, 20, 28, 31–33], some models just used point estimates for incidence as starting values [34, 35]. Model validation with data that was not used for calibration was only performed in two MC studies [26, 28].

**3.3.4 Vaccines.** Multiple studies have modeled the impact of vaccines against meningococcal serogroup B infections, especially since the vaccines' approval for use that started in the early 2010s for different countries and age groups.

**3.3.5 Vaccine effectiveness.** Vaccine effectiveness was estimated by vaccine efficacy alone or in combination with vaccine strain coverage and vaccine uptake. Vaccine efficacy against disease was high: 78 to 95% [21, 26, 27, 32, 35–37] and lower against carriage: 20 to 30% (exploring ranges up to 60%) [27, 36, 37]. The strain coverage was assumed to be between 66 and 90% [37], and uptake or vaccine coverage decreased with age, from around 90% in infants [21, 32, 35], to 60 to 75% in school children [27, 36], to 30% in adolescents outside school [36].

**3.3.6 Vaccine duration.** The duration of vaccine protection was assumed to be age dependent. A population up to 1 year of age with three or four doses only had a vaccine protection of 18 to 38 months [23, 26–29, 31, 34, 37]. Adolescents with one or two doses around the age of 14 were assumed to have a longer protection of 8 to 10 years [26, 27, 35, 37]. However, there were studies assuming longer protection for younger [21, 38] or shorter protection for older age groups [23]. Waning of protection was modeled as either a constant annual waning (e.g., in ordinary differential equation, ODE, type models) [26, 33, 34], or as a combination of a constant protection level during the protective period followed by a waning process [24, 35].

**3.3.7 Vaccine impact.** In the short term (up to about five years), the vaccine can lead to a reduction of cases of 5 to 27% [13, 20, 29, 31, 33, 36], with higher reductions in certain populations like small children (40–46%) [26, 32] or college students (63%) [23]. In the longer term (10 years to lifelong), the herd effect of vaccinations can lead to a higher reduction of cases of around 25 to 60% [13, 20, 29, 36]. Some models, however, do not include dynamic disease transmission and thus no vaccination herd effect, leading to lower long-term reductions [21, 22, 34].

**3.3.8 AMR.** Meningococcal AMR is still very rare for first-line antibiotics albeit sporadic reports of reduced susceptibility against cefotaxime, ceftriaxone or rifampicin, and increasing resistance to penicillin globally [39]. Limited efforts seem to have been dedicated to this problem, and only a single modeling study looking into AMR (Penicillin G resistance) in *N. meningitidis* was found [40].

**3.3.9 Health economics.** S1 Table in S1 File summarizes the evidence on the 11 studies we identified which investigated the cost-effectiveness of meningococcal serogroup B vaccination. Ten of the studies had very high incremental cost-effectiveness ratios (>£100,000 per QALY) suggesting, in most countries, that an MC vaccine would struggle to be deemed cost-effective. Methodological approaches varied considerably although key consistencies were the adoption of an effective life-time time horizon, the modeling of multiple alternative vaccination strategies, and the inclusion of the costs and harms of long-term sequelae associated with MC. Key drivers of cost-effectiveness was the prevalence of disease, the cost of the vaccine, the type of sequelae included, the use of QALY-scaling factors, and the discount rate. In most studies, the vaccine price would have had to be very low (<£10) to be considered cost-effective.

## 3.4 Gonococcal (GC) studies

**3.4.1 Model approaches.** GC studies used a greater variety of modeling approaches than MC studies, ranging from ordinary and partial differential equation systems to individual- or population-based Markov, to network models. Different approaches, often novel rather than built on previously published models, were used to account for the transmission in populations with non-random mixing.

**3.4.2 Model structure.** In general, all studies used an SIS approach, with infected individuals returning to the susceptible population after treatment or through natural clearance. As it has been shown that recovered individuals can be re-infected after short periods [41], the studies did not account for an immune state with the exception of the work by Duan et al. who used a very short immune period of only 3.5 days [42]. The infected state was divided into symptomatic and asymptomatic carriers in all studies.

**3.4.3 Site-specific modeling.** Site-specific infection dynamics are especially important for MSM. Here, gonococcal infections can occur in three sites: in the pharynx, urethra, and rectum, and a few studies take this into account [42–46]. In this case, models had to include site-specific transmission routes and infection rates, and calibrating transmission parameters to site-specific prevalence showed that the risk of infection is higher for the receiving partner

[45]. Other studies on MSM modeled GC infections in individuals rather than anatomical sites but exhibit similar dynamics to the site-stratified ones (compare e.g. [46] and [47]).

**3.4.4 Setting and population.**   GC modeling studies on vaccination or AMR only focus on developed countries (Western Europe, USA and Canada, Australia).Here, they often concentrated on certain risk groups, such as MSM (14), heterosexuals (9), sex workers (1) or indigenous people (1). Only three GC studies [47–49] looked at both the MSM and heterosexual population. However, they used separate models without any spill over for the two populations. In addition, the populations in GC studies were often stratified into high and low risk groups by their sexual activity, following the work on core groups in gonorrhoea transmission [50].

While early works were based on purely theoretical populations [17, 18, 51], more recent models were parameterized similar to the MC studies, mostly to national population and infection data [49, 52–60], some used averaged European and global prevalence estimates [27, 61]. Model validation with data that was not used for calibration was performed in three GC studies [42, 56, 58].

**3.4.5 AMR.**   All AMR studies identified in this search were for gonococcal infections, reflecting the change of first line antibiotics [62] from penicillin pre-1990s [17] to Ciprofloxacin in the 1990s [47], Cefixime in the 2000s [58], Azithromycin in the 2010s [49], to Ceftriaxone, which is currently used in the UK [63].

Here, the spread of AMR was modeled by using a susceptible and a resistant GC strain for infection [44, 47, 55, 64], multiple strains with different degrees of antibacterial susceptibility [49, 52], or multiple strains, each with a resistance against a different antibiotic [54, 65]. Model structures with and without co-infection of multiple such strains were compared by Turner and Garnett [18].

AMR cases were either imported [59] or arose through treatment [55, 64]. Without a substantial fitness cost associated with AMR, the resistant strains outperformed the susceptible ones in all studies, leading to the spread of AMR.

The models show that the balance between using more antibiotics to treat people and using less antibiotics to prolong their lifespan will be key in the future. Without treating less, the lifespan of antibiotics could be extended with AMR point-of-care testing [47, 48, 63, 65], combination therapy [53, 54, 66], different screening frequencies [55, 56], vaccines [52], or focus the screening on core groups [42, 46, 64, 67].

**3.4.6 Vaccines.**   We found ten studies modeling GC vaccinations. All ten studies looked at hypothetical vaccine benefits: they all screened the potential ranges for uptake, effectiveness (10 to 100%), and protection duration (1 to 20 years) in different scenarios or with sensitivity analyses.

The impact on the population was accordingly depending on the given scenario. The reduction of cases was around 10 to 40% for the heterosexual population [57, 61, 68, 69] and 45 to 66% for MSM [45, 52, 59, 70]. As for MC models, the longer the chosen study horizon the higher the reduction of cases due to vaccination herd effects.

**3.4.7 Health economics.**   S2 Table in S1 File summarizes modeling studies for gonococcal infection. There were three studies focusing on gonococcal infections [54, 68, 70] with the first two considering antimicrobial residencies. Only two studies focused on vaccination to prevent gonococcal [68, 70]. Régnier & Huels [68] used a Markov-based model to explore the cost-effectiveness of a hypothetical vaccine with differing effectiveness rates when vaccinating adolescents in the USA. They model men and women separately. Long-term sequelae associated with a gonococcal infection for women include ectopic pregnancy; chronic pelvic pain and infertility all of which have an impact on patient utility values and health care costs. For men, sequelae include urethritis, epididymitis, and an increase in the risk of HIV infection. Vaccination has a substantial impact on reducing infections, health, and costs which all result in a low

value-based price for the hypothetical vaccine, i.e., implying a potential vaccine is likely to be cost-effective. Important parameters driving the vaccine value related to the reduction in risk of HIV infection associated with fewer infections and a reduction in the number of sequelae occurring in women.

Whittles et al. [70] use an ODE model to explore the impact of vaccination on MSM in England. They model four different scenarios–Vaccination before entry (VbE), Vaccination on diagnosis (VoD), Vaccination on attendance (VoA) and Vaccination according to risk (VaR). They find the hybrid strategy of VaR to be the most cost-effective, leading to an overall reduction in costs (at £18 per dose) and a reduction in cases versus no vaccination. At a vaccine price of £85, VaR would likely be cost-effective at threshold of £30,000 per QALY. Whittles et al. do not model infection in women or associated sequelae, neither does it model long-term sequelae in MSM.

## 3.5 Quality evaluation

We performed a quality evaluation of all 52 included modeling studies. Of these studies, 30 were rated positive in at least seven of the nine categories (while some of these still failed to provide essentials such as a mathematical description of the model). All studies were judged to have reasonable model structure and assumptions with sufficient description of the transmission processes except for one study that referred to other work for the description (51/52). A total of 15 studies failed to provide a full mathematical description of the model, with the rest having equations either in the method's section or in the supplementary material. Most studies (44/52) performed some sort of model calibration with varying degrees of detail. 41/52 studies tested the influence of parameters in either parametric sensitivity or uncertainty analyses, or both. A structural sensitivity analysis in which different model types or model structures were compared, was only performed in 7/52 studies. Only 5/42 of the models validated their results with internal or external data, but all were judged to have face validity. In 31/52 studies, the authors declared some sort of conflict of interest, ranging from minor funding received by one or two authors (8/52) to all authors working for a vaccine-producing company (8/52).

## 4. Discussion

The incidence of gonorrhoea has increased year on year in Europe and the US before the SARS-CoV-2 pandemic and current numbers are the highest in decades [75–77]. While a large proportion of these infections show resistance to specific antibiotics [77], reduced susceptibility against the first-line antibiotic Ceftriaxone is still relatively low [78, 79]. More worryingly, multi-drug resistant (MDR) and extensive drug resistant (XDR) gonorrhoea are fast emerging in other parts of the world and can spread after importation [80]. The ability of *N. gonorrhoeae* to develop resistance to antibiotics has led to relatively early modeling efforts in this field, e.g., with analyzing the resistance of GC against penicillin [17]. However, while this field gains more and more traction now that gonorrhoea has developed resistance to all classes of antibiotics recommended for treatment, modeling of gonococcal AMR is still identified as one of most understudied AMR topics given its urgency [81].

In general, gonorrhoea modeling has been used to understand transmission dynamics and treatment scenarios, largely influenced by the work of Hethcote and Yorke who introduced core groups with a higher rate of partner changes [50]. However, since the recent observations of MenB vaccine effectiveness against gonorrhoea, there is a growing number of modeling papers in this field too. This is comparable with vaccination modeling for *N. meningitidis*: previously, models have been used to inform public health actions and vaccination campaigns against serogroups A, C and W in the African meningitis belt (e.g. [82]) and other parts of the

world (e.g. [19]). Serogroup B, however, has only become the subject of mathematical modeling in more recent years, especially with the introduction of specific vaccines like MeNZB, Trumenba and Bexsero. This led to the modeling of serogroup B meningococcal disease to inform vaccination policies and programmes in the 2010s, and the same is slowly starting with gonorrhoea, where models are used to analyze strategies for selected populations at risk. Given this recent increase in modeling approaches to *Neisseria* infections and their implications for treatment and vaccine strategies, it was necessary to review the literature so that future modeling studies have an overview on used assumptions, model approaches and research gaps. In this review, we found a broad range of model types used, with deterministic dynamical model and stochastic Markov model types dominating for both MC and GC infections. While both infections were modeled following a susceptible-infected-recovered/vaccinated-susceptible transmission cycle, MC models stratified the population by age groups whereas GC models stratified by sexual activity risk groups, each with according contact matrices for the respective *Neisseria* transmission.

In 2019, the WHO convened a multidisciplinary international group of experts to understand the potential health, economic and social value of gonococcal vaccines and to describe an ideal set of product characteristics for such a vaccine [83, 84]. The group identified that the overall strategic aim for a vaccine should be to: a) reduce the negative impact of infection on health outcomes and b) reduce the threat of gonococcal antimicrobial resistance (AMR). In the short-term, a reduction in the negative health consequences was deemed to be the priority with a particular focus on reducing the impact on women who tend to have the most severe sequelae [85], whereby an infection can cause pelvic inflammatory disease, infertility, chronic pelvic disease, and ectopic pregnancy. The health economics perspective sought to focus in particular detail on the negative consequences associated with infection and how these had been conceptualized in the existing modeling literature. We found only two studies that investigated the cost-effectiveness of vaccination to prevent gonococcal infection but eleven studies that investigated the cost-effectiveness of vaccination for meningococcal disease. In general, the meningococcal studies went to great lengths to integrate the consequential impact of infection on sequelae and the knock-on patient outcomes and costs. However, despite the inclusion of these potential sources of value, vaccination for MC was often unlikely to be cost-effective because it required significant investment in vaccination to prevent very serious but very rare events. By contrast, the sequelae incorporated in the GC models were generally limited, unjustifiably so, particularly in terms of the impact of GC infection on women, which can be considerable. Yet both studies did demonstrate the potential for a cost-effective vaccine even when only partially incorporating the value of a potential vaccine.

The attempts to model vaccination strategies against GC show that empirical studies in the lab or clinical trials are necessary to get a better picture of MenB vaccine characteristics against GC infections. Randomized-controlled clinical trials on the effectiveness of the vaccine are currently under way in the US and Thailand for heterosexuals [86], and in Hong Kong [87] and Australia [88] for MSM. In addition, another gonorrhoea vaccine was recently fast tracked in the US [89], and is now entering a phase 2 trial [90]. More detailed information on vaccine characteristics will in turn help inform cost-effectiveness analyses looking at the general population or certain risk groups. These can then be used to inform public health action and policies, comparable to how cost-effectiveness studies of MenB vaccination against meningococcal disease have shaped vaccination strategies in several countries [13, 32]. In fact, following the analysis of Whittles et al. and Looker et al. [57, 70], the British Joint Committee on Vaccination and Immunisation (JCVI) has now recommended the use of Bexsero for those who are at greatest risk of gonococcal infections in the UK [91].

As our review has shown, there are already a good number of options for model structures and assumptions available to study *Neisseria* infections. Not relying on a single approach is

very useful to check the influence of assumptions. Especially for GC, lots of models have been developed independently (albeit mostly influenced by some early work on sexual networks) which contributed to the diversity of approaches. Nevertheless, we identified four key gaps in the modeling work on strategies against *Neisseria* infections and AMR development:

1. Modeling vaccination effect on both GC and MC infections:
   Meningococcal serogroup B vaccines prevent both MC and GC disease. So far, any vaccination strategies have only been studied for each disease separately. Including both infections could show the combined positive public health impact and give better cost-effectiveness estimates.

2. Combination of MSM and heterosexual population for GC studies:
   As with all STIs, there is spill over from the heterosexual to the MSM community and vice versa. A realistic model with both groups could, for example, account for the non-linear effects that a vaccination campaign in one group can have on the other.

3. Health economics including sequelae specific to women for GC studies:
   The current cost-effectiveness studies on gonorrhoea vaccines focused on MSM and thus only accounted for a limited number of sequelae in women. However, sequelae from gonococcal infections in women can be very severe, so that their inclusion could make non-targeted vaccine programmes much more cost effective.

4. Settings in low- and middle-income countries for GC studies:
   We have not found any GC modeling studies on AMR or vaccination strategies set in a developing country. GC is a common disease all over the world and it is necessary to support public health systems in developing countries with studies on how to tackle AMR. This would also be beneficial to richer countries, as, for example, most Ciprofloxacin-resistant cases are imported from the Asia-Pacific region to the UK [78].

While we did look at different risk groups like MSM and the heterosexual population for our review, we did not specifically check for vaccination impacts on different age groups. Studies suggest that especially vaccination duration differs significantly by age, so this could be done in a meta-analysis for either meningococcal or gonococcal infections. Using multiple grey literature data bases and relative broad search terms in the screening process yielded a wide variety of *Neisseria* modeling approaches. This inevitably also led to the inclusion of studies that were not directly aligned with our question but still relevant in the field. The used assessment tool developed by Lo et al. [16] should thus be seen as an indicator of how useful the studies are for our purpose, rather than of their quality. Nevertheless, the assessment emphasizes that a clear documentation and the inclusion of uncertainty analyses should be the standard when modeling infectious disease scenarios that should influence public health action. A final limitation of our systematic review is that its protocol was not registered prospectively with PROSPERO. The literature search had already started before we thought about registering and thus it was not possible anymore, but for comparison of protocol changes and to avoid possible duplication efforts, it should have been done. For transparency, we have uploaded the original draft of the protocol in S6 File.

In conclusion, George Box's aphorism, 'All models are wrong, but some are useful,' aptly frames the two disease areas studied in this literature review. For MC, we found that most models investigating the cost-effectiveness of vaccination went to great lengths to incorporate the potential value of avoiding the debilitating, life-limiting, and devastating sequelae of the disease. These models often included detailed considerations of the quality-of-life impacts during and after the acute disease episode, long-term health consequences such as scarring,

paralysis, and neurological disorders, and even indirect costs such as legal claims against healthcare systems. Yet, despite these comprehensive analyses, the upfront cost of mass vaccination against MC was often not deemed to be cost-effective due to the relatively low incidence of these severe occurrences.

By contrast, our review found that for GC, existing models predominantly focus on high-risk populations, such as men who have sex with men or heterosexual men. This is despite the WHO's expert group in 2019 emphasizing the need for a gonococcal vaccine to primarily reduce the health consequences of infection, especially in women, who are disproportionately affected. Many women with gonorrhea are asymptomatic, potentially leading to chronic infections without treatment, resulting in pelvic inflammatory disease, infertility, chronic pelvic pain, and ectopic pregnancy. This oversight in modeling represents a significant limitation in current strategies, failing to fully capture the value of vaccination approaches. Yet, in the few studies that do investigate the cost-effectiveness of GC vaccination, even without adequately considering the impact on women, vaccination still appears to be potentially cost-effective. Future modeling studies should always seek to fully characterize the potential for spillovers across populations, such as into women, where the short and long-term cost-consequences are likely to be an important part of the whole decision-making picture.

The future for vaccination against *Neisseria* infections looks promising though: for MC, a pentavalent MenABCWY vaccine for individuals aged 10 to 25 has recently been approved in the USA [92], and could increase MC vaccination coverage for all five serogroups. This vaccine uses Trumenba for the B component and thus its effectiveness against gonorrhoea infection is yet unclear. Another pentavalent vaccine currently in phase III clinical trials [93] uses Bexsero for the B component and could thus also offer some protection against gonorrhoea should it be approved. That said, vaccines specifically against GC are also under development, including a vaccine currently being developed by INTRAVACC [94], and the aforementioned vaccine by GSK [90] that in turn might offer some level of cross protection against MenB.

## Supporting information

**S1 File. Additional figures and tables.**
(DOCX)

**S2 File. Data extraction.**
(XLSX)

**S3 File. Assessment of included studies.**
(XLSX)

**S4 File. PRISMA checklist.**
(DOCX)

**S5 File. Study protocol.**
(DOCX)

**S6 File. Original draft of study protocol.**
(DOCX)

**S7 File. Literature search.**
(XLSX)

## Author Contributions

**Conceptualization:** Soeren Metelmann, Segun Oke, Lorenzo Pellis, Ian Hall.

**Formal analysis:** Soeren Metelmann, Alexander Thompson, Anna Donten, Segun Oke.

**Funding acquisition:** Alexander Thompson, Roberto Vivancos, Valerie Decraene, Lorenzo Pellis, Ian Hall.

**Investigation:** Alexander Thompson, Suzy Sun, Ray Borrow.

**Methodology:** Soeren Metelmann.

**Supervision:** Ray Borrow, Roberto Vivancos, Valerie Decraene, Lorenzo Pellis, Ian Hall.

**Validation:** Soeren Metelmann.

**Visualization:** Soeren Metelmann.

**Writing – original draft:** Soeren Metelmann, Alexander Thompson.

**Writing – review & editing:** Soeren Metelmann, Alexander Thompson, Anna Donten, Segun Oke, Suzy Sun, Ray Borrow, Feng Xu, Roberto Vivancos, Valerie Decraene, Lorenzo Pellis, Ian Hall.

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
