## [Decision Letter · Decision Letter 0]

27 Aug 2024

PONE-D-24-24851A systematic review to identify research gaps in studies modeling vaccination strategies against Neisseria infectionsPLOS ONE

Dear Dr. Metelmann,

Thank you for submitting your manuscript to PLOS ONE. After careful consideration, we feel that it has merit but does not fully meet PLOS ONE’s publication criteria as it currently stands. Therefore, we invite you to submit a revised version of the manuscript that addresses the points raised during the review process.

The referees have found your manuscript suitable for publication but raised some concerns regarding clarity. Please, consider the remarks by Reviewer #2 in your revision, by expanding the Introduction and Discussion about the scope of your review and the conclusions you derived from it. 

We look forward to receiving your revised manuscript.

Kind regards,

Michele Tizzoni

Academic Editor

PLOS ONE

Journal Requirements:

   "I have read the journal's policy and the authors of this manuscript have the following competing interests:

RB performs contract research on behalf of UKHSA for GSK, Pfizer, and Sanofi. RV has received research funding for PHE from GSK and Gilead Sciences in the past. All other authors have declared that no competing interests exist."

We note that you received funding from a commercial source: GSK and Gilead Sciences

5. Please ensure that you refer to Figure 3 in your text as, if accepted, production will need this reference to link the reader to the figure.

Reviewers' comments:

Reviewer's Responses to Questions

**Comments to the Author**

1. Is the manuscript technically sound, and do the data support the conclusions?

Reviewer #1: Yes

Reviewer #2: Partly

2. Has the statistical analysis been performed appropriately and rigorously? 

Reviewer #1: Yes

Reviewer #2: N/A

3. Have the authors made all data underlying the findings in their manuscript fully available?

Reviewer #1: Yes

Reviewer #2: Yes

4. Is the manuscript presented in an intelligible fashion and written in standard English?

Reviewer #1: Yes

Reviewer #2: Yes

5. Review Comments to the Author

Reviewer #1: This systematic review focuses on studies on vaccination strategies against Neisseria incidence and antimicrobial resistance. The primarily aim is to identify gaps in research and provide suggestions for future research, including the combination of both meningococcal and gonococcal infections in studies to provide better estimates for vaccine benefits. The review is well written and the topic is relevant.

I have a few minor comments below:

- Typo in Results section paragraph 3.2 where the sentence "Error! Reference source not found." is included.

- Paragraph 3.3.1: Could the authors provide a brief definition of difference between deterministic ODE and stochastic Markov models?

- Table 2: there is a typo for the year of publication which is 2016 and not 2006

- Point 3 in the Discussion section: should be sequelae instead of sequalae?

- Figure 2 is not readable and same for Figure 3

Reviewer #2: This is a very interesting topic for a review. However the paper has major limitations. First of all, the paper only focuses on Men B vaccination and its effect on N. gonorrhoeae. The authors should make the goal of their study more clear in the title and abstract, as they explicitly exclude all studies on meningococcal conjugate vaccine that are not type B. Second, the result section should include more details on the studies included in the review: what is the data used (did they use disease incidence records? were the studies mostly free simulations or did they provide inference based on real data?) , what was the impact of vaccination at the population level? The authors need to describe more in details these results for this review to be useful for researchers.

6. PLOS authors have the option to publish the peer review history of their article (what does this mean?). If published, this will include your full peer review and any attached files.

Reviewer #1: No

Reviewer #2: No

---

## [Author Response · Author response to Decision Letter 0]

25 Nov 2024

Please see all responses to comments in the attached file "response.docx".

---

## [Editor Report · Decision Letter 1]

9 Dec 2024

A systematic review to identify research gaps in studies modeling MenB vaccinations against Neisseria infections

PONE-D-24-24851R1

Dear Dr. Metelmann,

We’re pleased to inform you that your manuscript has been judged scientifically suitable for publication and will be formally accepted for publication once it meets all outstanding technical requirements.

Kind regards,

Michele Tizzoni

Academic Editor

PLOS ONE

Additional Editor Comments (optional):

After reading the revised version of the manuscript and the response to the referees, I am convinced the authors addressed all comments by the referees and the paper can be accepted for publication in PLOS ONE.
---

## [Editor Report · Acceptance letter]

18 Dec 2024

PONE-D-24-24851R1 

PLOS ONE

Dear Dr. Metelmann, 

I'm pleased to inform you that your manuscript has been deemed suitable for publication in PLOS ONE. Congratulations! Your manuscript is now being handed over to our production team.

Kind regards, 

on behalf of

Dr. Michele Tizzoni 

Academic Editor

PLOS ONE